# Automated triaging of head MRI examinations using convolutional neural networks

**David A. Wood**[1]                                    DAVID.WOOD@KCL.AC.UK
**Sina Kafiabadi**[2]                                   SKAFIABADI@NHS.NET
**Aisha Al Busaidi**[2]                                 AYISHA.ALBUSAIDI@NHS.NET
**Emily Guilhem**[2]                                    EMILY.GUILHEM@DOCTORS.ORG.UK
**Antanas Montvila**[2]                                 MONTVILA.ANTANAS@GMAIL.COM
**Siddharth Agarwal**[1]                                SIDDHARTH.1.AGARWAL@KCL.AC.UK
**Jeremy Lynch**[2]                                     JEREMY.LYNCH@NHS.UK
**Matthew Townend**[3]                                  MATTHEW.TOWNEND@WWL.NHS.UK
**Gareth Barker**[4]                                    GARETH.BARKER@KCL.AC.UK
**Sebastian Ourselin**[1]                               SEBASTIEN.OURSELIN@KCL.AC.UK
**James H. Cole**[4,5]                                  JAMES.COLE@UCL.AC.UK
**Thomas C. Booth**[1,2]                                THOMAS.BOOTH@KCL.AC.UK

[1] *School of Biomedical Engineering, King's College London*

[2] *King's College Hospital*

[3] *Wrightington, Wigan and Leigh NHSFT*

[4] *Institute of Psychiatry, Psychology & Neuroscience, King's College London*

[5] *Centre for Medical Image Computing, Dementia Research, University College London*

**Editors:** Under Review for MIDL 2021

## Abstract

The growing demand for head magnetic resonance imaging (MRI) examinations, along with a global shortage of radiologists, has led to an increase in the time taken to report head MRI scans around the world. For many neurological conditions, this delay can result in increased morbidity and mortality. An automated triaging tool could reduce reporting times for abnormal examinations by identifying abnormalities at the time of imaging and prioritizing the reporting of these scans. In this work, we present a convolutional neural network for detecting clinically-relevant abnormalities in $T_2$-weighted head MRI scans. Using a validated neuroradiology report classifier, we generated a labelled dataset of 43,754 scans from two large UK hospitals for model training, and demonstrate accurate classification (area under the receiver operating curve (AUC) = 0.943) on a test set of 800 scans labelled by a team of neuroradiologists. Importantly, when trained on scans from only a single hospital the model generalized to scans from the other hospital ($\Delta$AUC $\leq$ 0.02). A simulation study demonstrated that our model would reduce the mean reporting time for abnormal examinations from 28 days to 14 days and from 9 days to 5 days at the two hospitals, demonstrating feasibility for use in a clinical triage environment.

## 1. Introduction

Magnetic resonance imaging (MRI) is fundamental to the diagnosis and management of a range of neurological conditions (Atlas, 2009). For many of these (e.g., acute stroke, brain tumour, haemorrhage), early detection can lead to better outcomes by increasing the likelihood that a patient will respond positively to treatment (Adams et al., 2005)(Kidwell et al.,

2004). In recent years, however, a growing demand for head MRI examinations, along with a global shortage of radiologists, has led to an increase in the time taken to report head MRI scans around the world (Bender et al., 2019). In the UK, for example, the reporting time for out-patient head MRI scans has increased every year since 2012 (NHS, 2019), with only 2% of radiology departments currently able to fulfill their imaging reporting requirements within contracted hours (RCR, 2017). Given the increasingly aging global population (UN, 2019), as well as additional backlogs created as a result of resource re-allocation in radiology departments during the global COVID-19 pandemic (Sangwaiya and Redla, 2020), reporting times are likely to continue to increase in the coming years, putting a growing number of patients at risk.

One solution to reduce reporting times for abnormal head scans is to develop an automated triage tool to identify abnormalities at the time of imaging and prioritize the reporting of these scans. Convolutional neural networks (CNN) show considerable promise for this purpose, having achieved remarkable success on a range of medical imaging tasks (McKinney et al., 2020)(Kamnitsas et al., 2016)(Ding et al., 2018). However, a bottleneck to the development of a CNN-based tool for triaging routine hospital head MRI examinations is the difficulty of obtaining large, clinically-representative labelled datasets to enable supervised learning (Hosny et al., 2018)(Wood et al., 2020a). A number of strategies have been proposed to deal with this limited availability of labelled training data. Several studies have demonstrated a hybrid approach to neurological abnormality detection by combining deep learning with atlas-based image processing and Bayesian inference (Rauschecker et al., 2020)(Rudie et al., 2020). Overall, however, unsupervised approaches have attracted the most attention, following the pioneering work of (Schlegl et al., 2017). The basic idea common to these studies is to train a generative model (e.g., a generative adversarial network (GAN) (Baur et al., 2020a)(Han et al., 2020), a variational autoencoder (VAE) (Chen and Konukoglu, 2018)(You et al., 2019)(Kobayashi et al., 2020)(Baur et al., 2020b)(Zimmerer et al., 2018)(Zimmerer et al., 2019) or a combination of the two (Baur et al., 2018)) to learn the manifold of normal anatomical variability, and at test time detect abnormalities by looking for outliers in either the latent feature space or the reconstruction loss. Importantly, only healthy images are needed for model training and these can be obtained from open-access research databases such as the Alzheimer's Disease Neuroimaging Initiative (ADNI) (Petersen et al., 2010) or the Human Connectome Project (HCP) (Van Essen et al., 2013).

Despite showing considerable promise, several important limitations to these unsupervised studies can be identified. In the majority of cases, images had undergone computationally expensive pre-processing steps such as bias-field correction, skull-stripping and spatial registration (Chen and Konukoglu, 2018)(Han et al., 2020)(Baur et al., 2018)(You et al., 2019)(Baur et al., 2020b)(Baur et al., 2020a)(Pawlowski et al., 2018) which limits real-time clinical utility and, in the case of skull stripping, precludes the detection of important extracranial abnormalities (Fig. 1) (e.g., orbital and sinonasal masses). Furthermore, model evaluation was often performed using datasets containing only a single class of abnormality (e.g., brain tumours (Chen and Konukoglu, 2018)(You et al., 2019)(Han et al., 2020)(Zimmerer et al., 2018)(Zimmerer et al., 2019), or white matter lesions (Baur et al., 2018)(Baur et al., 2020a)(Baur et al., 2020b)), whereas a triage tool needs to detect a range of abnor-

malities, including subtle but clinically important vascular abnormalities (e.g., subarachnoid haemorrhage or venous sinus thrombosis). Finally, in a number of studies training was performed exclusively with images of healthy *young* adults (i.e., 22 - 35 years) (Chen and Konukoglu, 2018)(Pinaya et al., 2019)(Zimmerer et al., 2019)(Zimmerer et al., 2018). This precludes learning the manifold of normal anatomical variability in the aging brain, particularly the appearance of small focal areas of increased signal intensity on $T_2$-weighted images scattered throughout the cerebral white matter which have been estimated to occur in $> 90\%$ of patients between 60-90 years old (LeMay, 1984a)(De Leeuw et al., 2001), as well as involutional atrophic changes (Golomb et al., 1993). A clinically useful triage system must be able to distinguish between changes which in a hospital setting are considered 'normal for age' and those considered 'excessive for age' (Fig. 1) and it is likely that models trained only on images of young (i.e. $\leq 65$ years (LeMay, 1984b)) healthy adults would fail to make this distinction.

In this work, we pursue a different strategy to overcome the limited availability of labelled head MRI scans. Rather than attempting to learn without labels, we instead sought to convert archived hospital examinations into a large labelled dataset suitable for supervised learning by deriving labels from the accompanying radiology reports using a validated neuroradiology report classifier. A benefit of using large-scale historical clinical data is that the full gamut of abnormalities likely to be encountered in a real-world hospital setting are seen during training. Furthermore, findings which are 'normal for age' can be distinguished from those which are 'abnormal for age' simply by including the patient age as input to the model, since the accompanying radiology reports (from which image labels are derived) reliably make this distinction (Wood et al., 2020a).

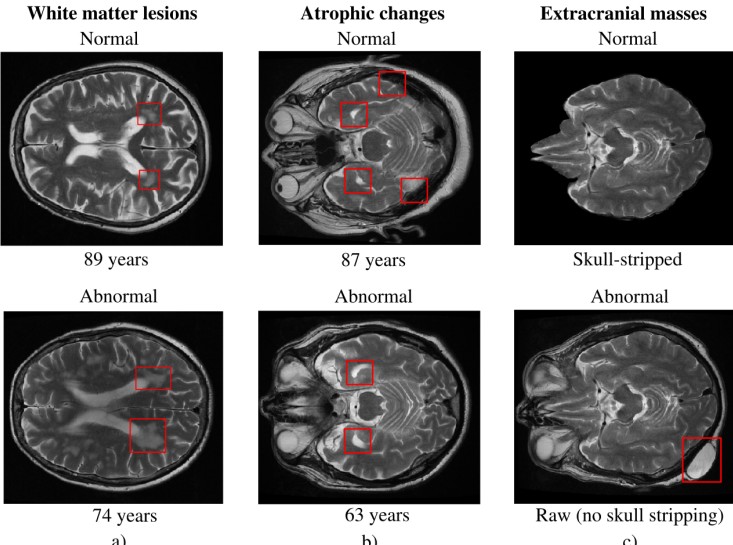

Figure 1: (a) Small foci of increased signal intensity on $T_2$-weighted images scattered throughout the cerebral white matter occur naturally in the aging brain (top) but are considered abnormal in younger patients (bottom). (b) Atrophic changes are part of normal aging (top), but can indicate early-onset neurodegenerative conditions in younger patients (bottom). (c) Important extracranial abnormalities (bottom) are missed when skull-stripping is performed (top). Abnormality detection systems suitable for clinical settings should distinguish between normal and abnormal changes in the aging brain, and detect extracranial abnormalities.

## 2. Methods

### 2.1. Data

All 54,115 adult ($> 18$ years old) axial $T_2$-weighted MRI head scans performed at King's College Hospital NHS Foundation Trust (KCH) and Guy's and St Thomas' NHS Foundation Trust (GSTT) between 2008 - 2019 were used in this study. The MRI scans were obtained on Signa 1.5 T HDX (General Electric Healthcare) or AERA 1.5T (Siemens), and were extracted from the Patient Archive and Communication Systems. The corresponding 54,115 radiology text reports produced by expert neuroradiologists were also obtained.

A subset of 5000 reports from KCH was randomly selected for annotation by 3 expert neuroradiologists in order to develop the neuroradiology report classifier (ALARM) described in (Wood et al., 2020b). Prior to report labelling, a complete set of clinically relevant categories of neuroradiological abnormality, as well as the rules by which reports were to be labelled, was developed (Appendix F). Broadly speaking, findings which would generate a downstream clinical intervention were labelled 'abnormal', as were those which would be referred for case discussion at a multi-diciplinary team meeting. ALARM achieved an AUC of 0.992 on a hold-out set of 500 manually-annotated KCH reports. However, differences in reporting styles could plausibly lead to poor performance ('domain shift') when classifying reports from an external hold-out set. To investigate this, 500 GSTT reports were randomly selected for annotation by the same 3 neuroradiologists. ALARM achieved an AUC of 0.990 on this external hold-out set of reports, demonstrating that it can be reliably used to label MRI examinations at both KCH and GSTT (Fig. S1, Table S1). Following this important validation step, ALARM was used to assign labels to all 43,754 axial $T_2$-weighted scans obtained from the two sites between 2008 - 2018 for computer vision model development, and to all 4861 scans obtained between 2018 - 2019 for use in a simulation study (Table S2). For computer vision model evaluation, a test set of 800 $T_2$-weighted scans with 'reference standard image labels' was generated by randomly sampling 40 examinations from each site for each year between 2008 - 2018. Two neuroradiologists labelled these scans as 'normal' or 'abnormal' applying the same framework used for report labelling - but interrogating the actual images. Importantly, this dataset contains more than 90 classes of morphologically distinct abnormalities. Further dataset information is provided in Appendix B.

### 2.2. Models

We trained (1) a baseline classification model, and (2) a classification model with an additional 'noise-correction' layer optimised for learning in the presence of label errors (Fig. 2). Both models utilize a 3D Densenet121 network (Huang et al., 2017)) for visual feature extraction, with the output of the final global average pooling layer concatenated with the patient's age and passed through a fully-connected layer (with softmax) to generate prediction probabilities for the two classes (for architecture details, see Appendix G).

Because our neuroradiology report classifier is not a perfect model (i.e. it achieves AUC $< 1$, Fig. S1), some small fraction ($\sim 5\%$) of the training images will be erroneously labelled 'normal' when in fact they should be labelled 'abnormal', and vice versa. Recent

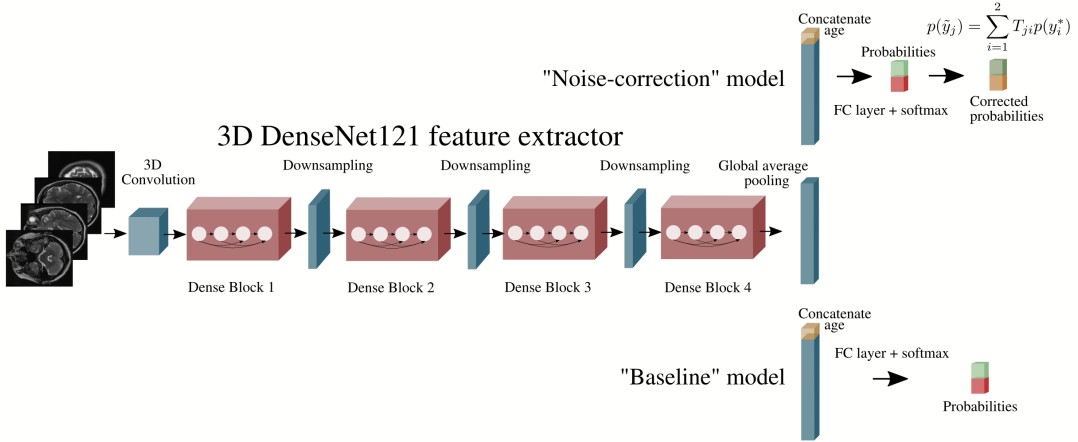

Figure 2: Baseline classification model and 'noise correction' classification model. Both networks perform visual feature extraction using a 3D Densenet121, and concatenate this with the patient's age in order to generate class probabilities. The 'noise-correction' model includes an additional layer which modifies the predictions during training to enable learning the true, rather than the noisy, label distribution.

studies have shown that this 'label noise' can significantly impact the performance of deep learning models (Zhang et al., 2017)(Karimi et al., 2020). Following (Patrini et al., 2017) and (Sukhbaatar et al., 2015), we seek to overcome label noise by adding a 'noise-correction' layer to our network. To motivate this, we note that the probability that a given image $\mathbf{x}$ with true (but unknown) label $y^*$ will be assigned a noisy label $\tilde{y}$ can be written as

$$p(\tilde{y} = j|\mathbf{x}) = \sum_i p(\tilde{y} = j|y^* = i)p(y^* = i|\mathbf{x}) = \sum_i T_{ji}p(y^* = i|\mathbf{x}), \tag{1}$$

where $T$ is a $2 \times 2$ 'transition matrix' with diagonal elements that specify the probability of correct labelling and off-diagonal elements which specify the probability of label 'flipping':

$$T = \begin{bmatrix} p(\tilde{y} = 0|y^* = 0) & p(\tilde{y} = 0|y^* = 1) \\ p(\tilde{y} = 1|y^* = 0) & p(\tilde{y} = 1|y^* = 1). \end{bmatrix} \tag{2}$$

In Eqn. 1, $p(y^*|\mathbf{x}; \theta)$ is the distribution which we desire our model to learn (i.e., the probability distribution of the true label, conditioned on input image $\mathbf{x}$), whereas $p(\tilde{y}|\mathbf{x}, \theta)$ is the distribution that is *actually* learned (e.g., by the baseline model) as a result of maximizing the cross entropy between the noisy labels $\tilde{y}$ and the model predictions. However, we can force the model to learn the true distribution $p(y^*|\mathbf{x}; \theta)$ by weighting the predicted probabilities during training by the corresponding elements of $T$ implied by Eqn. 1 - an operation which can conveniently be recast as a matrix multiplication between the $2 \times 2$ matrix $T$, and the $2 \times 1$ softmax output. At test time, when reference standard image labels are available, $T$ is set to the identity matrix ($I_2$) to enable predictions on the basis of $p(y^*|\mathbf{x}; \theta)$.

In general, $T$ is unknown and must be learned as part of model training. As described in (Sukhbaatar et al., 2015), however, this often results in $T$ converging to $I_2$, in which case the baseline and 'noise-corrected' models are identical. In the case of label errors resulting from imperfect text classification, however, an accurate estimate of $T$ is provided by the confusion matrix which is typically generated as part of NLP model validation (Table S1). A novel contribution of our work is to show the efficacy of this 'estimated loss correction' procedure when training models on medical image datasets labelled using NLP.

## 3. Experiments

A number of models were trained using different subsets of the available NLP-labelled data. In each case, the data were split into training (80%) and validation (20%) sets, ensuring that no patient appearing in the training set appeared in the validation set. Final model evaluation was always performed on a test set of images of unseen patients with reference standard labels assigned by neuroradiologists on the basis of manual image inspection. Our DenseNet model was based on the Project MONAI (MONAI, 2020) implementation, and all modelling was performed with PyTorch 1.7.1 (Paszke et al., 2019). Minimal image pre-processing was performed; all raw axial $T_2$-weighted images were re-sampled to a common voxel size $(1 \text{ mm}^3)$, and then resized to $(120 \times 120 \times 120)$. We applied histogram standardization to each image, but no spatial registration, bias-field correction or skull stripping was performed. ADAM optimizer (Kingma and Ba, 2017) was used with an initial learning rate 1e-4 which was reduced by a factor of 10 after every 5 epochs without validation loss improvement. Training was repeated 5 times for each model using different training/validation data splits in order to generate confidence intervals (test sets remained fixed). DeLong's test (DeLong et al., 1988) was used to determine the statistical significance of differences in ROC-AUC, and occlusion sensitivity was used to interrogate model decisions.

To quantify the impact that our model would have in a real clinical setting, we performed a retrospective simulation study using all out-patient examinations performed at KCH and GSTT between 1/1/2018 - 31/12/2018 to determine what would have happened if our model had been used to suggest the order in which head MRI examinations were reported. Full details of the simulation are presented in Appendix C. We also conducted Patient and Public Initiative (PPI) meetings to gauge the attitudes of patients, their families, and end-users (i.e. neuroradiology department personnel) to AI-assisted triage (Appendix D).

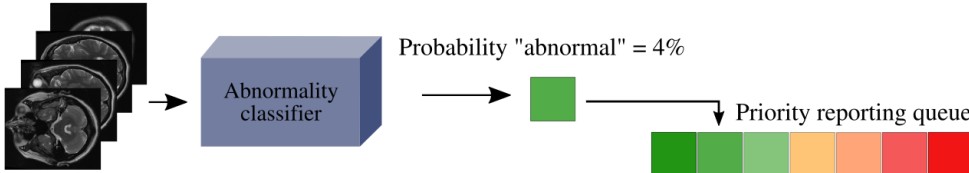

Figure 3: Our classifier can be used to suggest the order in which head MRI examinations are reported by inserting images in real-time into a dynamic reporting queue based on the predicted likelihood of being abnormal (shown) or on the predicted category and time spent in the queue (what we do in this study).

### 3.1. Results

Accurate classification (AUC > 0.9) was seen for both models for all training/testing combinations. However, 'noise-correction' led to a small but statistically significant improvement in all cases (p < 0.05)(Table 1). When trained on scans from only a single hospital the models generalized to scans from the other hospital ($\Delta$AUC $\leq$ 0.02) (Fig. 4). Occlusion analysis shows that true positive predictions are sensitive to salient image features (Fig. S2). Table 2 shows the impact that the best performing model (AUC = 0.943) would have had if it was used to suggest the order that examinations were reported. At both hospitals, the reduction in reporting times for abnormal examinations, as well as the increased reporting times for normal examinations, was statistically significant (p < 0.001) (Fig. 5).

| Train | | KCH | | | GSTT | | | Pooled | | |
|---|---|---|---|---|---|---|---|---|---|---|
| Test | | KCH | GSTT | Pooled | KCH | GSTT | Pooled | KCH | GSTT | Pooled |
| Model | Baseline | 0.921 | 0.909 | 0.915 | 0.903 | 0.918 | 0.912 | 0.925 | 0.920 | 0.922 |
| | Noise-corrected | 0.941 | 0.925 | 0.933 | 0.929 | 0.931 | 0.930 | 0.946 | 0.939 | 0.943 |

Table 1: Classification performance (AUC) for the baseline and 'noise-corrected' models. Both show accurate classification (AUC> 0.9), but 'noise correction' led to an improvement for all train/test splits (p < 0.05).

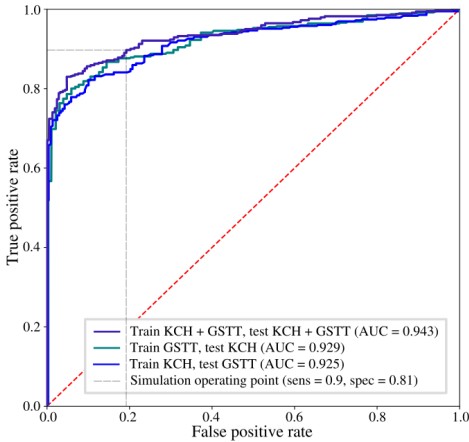

Figure 4: Receiver operating characteristic curve for the 'noise-correction' model (1) trained/tested using images from both sites (purple), (2) trained on KCH, tested on GSTT (teal), and (3) trained on GSTT, tested on KCH (blue). Operating point used for the simulation study is also shown (dotted grey).

| | | Time to report | |
|---|---|---|---|
| | | Normal | Abnormal |
| GSTT | Historical | 31 ± 21 days | 28 ± 22 days |
| | Our model | 36 ± 40 days | 14 ± 23 days |
| KCH | Historical | 10 ± 8 days | 9 ± 7 days |
| | Our model | 15 ± 21 days | 5 ± 7 days |

Table 2: Results of the retrospective simulation study, demonstrating the impact that our model would have on reporting times for abnormal scans at KCH and GSTT. Data are mean delay ± standard deviation.

## 4. Discussion

Our work builds on recent breakthroughs in natural language processing which have made it feasible to derive labels from radiology reports and assign these to the corresponding images (Vaswani et al., 2017)(Devlin et al., 2018)(Wood et al., 2020b), enabling the application of supervised learning at scale. Following (Annarumma et al., 2019)(Wang et al., 2020) and (Titano et al., 2018), we have put our model into clinical context through a retrospective simulation, demonstrating that it would reduce the reporting times of abnormal examinations at two real-world hospitals. A consequence of this is that the time to report normal examinations will be increased, and this may present issues for the few false negative errors which our model makes. Our team of neuroradiologists have determined that mistakes primarily occur ($\sim$ 88%) with findings which are most naturally described in terms of a

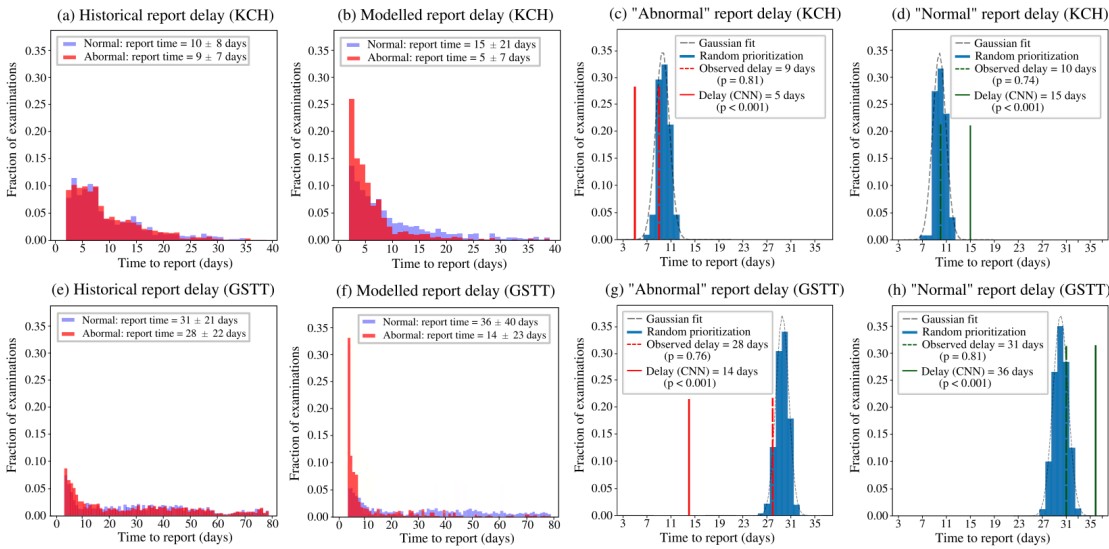

Figure 5: Retrospective simulation results for KCH (top) and GSTT (bottom). Historical reporting delays (a, e) are compared with what would have been observed if our model had been used to prioritize the reporting of abnormal scans (b, f) at the two sites. To test for statistical significance, the null hypothesis distribution was generated (c, d, g, h) by repeating the simulation 1000 times, assigning a random priority to each examination (blue). At both sites, a statistically significant ($p < 0.001$) reduction in reporting times for abnormal examinations (solid red) compared with what was observed historically (dashed red) was seen.

'spectrum', but which we have elected to binarize to enable supervised learning. For example, 'minor', 'mild' or 'modest' small vessel disease (SVD) is considered 'normal', whereas 'moderate' or 'severe' SVD is considered 'abnormal'. In most cases, our model was able to correctly classify SVD; however, equivocal cases (e.g., 'mild-to-moderate', which had been labelled 'abnormal' to encourage model sensitivity) were sometimes misclassified. Likewise, equivocal cases involving atrophy and enlarged perivascular spaces were sometimes misclassified. Given the degree of subjectivity involved (Fig. S3), these errors are highly unlikely to have a significant clinical impact. Nonetheless, as future work we plan to investigate the use of regression, rather than binary classification, to model these particular abnormalities.

A limitation of our model is that abnormalities which are not visualisable on $T_2$-weighted scans will not be detected. For example, microhaemorrhages and blood breakdown products are sometimes only visible on gradient-echo or susceptibility-weighted images. However, these sequences are not typically part of routine head examinations, so this is not a major issue in practice. A further limitation is that some abnormalities in our 'abnormal' category require more urgent intervention than others. As part of future work, we plan to develop a third category of 'emergency diagnoses' to finesse the triage process further. However, UK NHS hospitals require that all emergency MRI scans be reported within 24 hours. Therefore, the benefit of a third category in our healthcare system is likely to be modest.

## 5. Conclusion

In this work we have presented a head abnormality classifier trained on 43,754 $T_2$-weighted head MRI scans labelled using a neuroradiology report classifier, and demonstrated accurate classification on a test set of 800 scans containing over 90 classes of morphologically distinct abnormalities. We have shown that the model would reduce the time to report abnormal examinations at two UK hospitals, demonstrating feasibility as an automated triage tool.

## Acknowledgments

We thank Joe Harper, Justin Sutton, Mark Allin and Sean Hannah at KCH for their informatics and IT support, Ann-Marie Murtagh at KHP for research process support, and KCL administrative support - particularly from Alima Rahman, Denise Barton, James Bingham and Patrick Wong.

This work was supported by the Royal College of Radiologists, King's College Hospital Research and Innovation, King's Health Partners Challenge Fund, NVIDIA (through the unrestricted use of a GPU obtained in a competition), and the Wellcome/Engineering and Physical Sciences Research Council Center for Medical Engineering (WT 203148/Z/16/Z)

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

## Appendix A. Neuroradiology report classifier validation

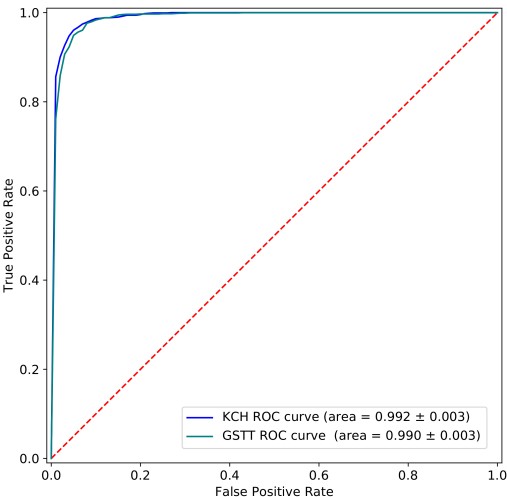

Figure S1: Reciever operating characteristic (ROC) curve for our neuroradiology report classifier (ALARM), described in (Wood et al., 2020b), evaluated on 500 radiology reports from both KCH (blue) and GSTT (teal) which had been manually labelled by a team of 3 neuroradiologists. The model generalized to reports at GSTT, despite being trained only on reports from KCH ($\Delta$AUC = 0.002), demonstrating that it can be reliably used to automate the labelling of MRI examinations at both sites.

|  |  | True label | | |
|---|---|---|---|---|
|  |  | Normal | Abnormal | Total |
| Predicted label | Normal | 428 | 29 | 457 |
|  | Abnormal | 33 | 510 | 543 |
|  | Total | 461 | 539 | 1000 |

Table S1: ALARM confusion matrix for a pooled (KCH and GSTT) test set of annotated radiology reports.

## Appendix B. Training and testing datasets

The UK's National Health Research Authority and Research Ethics Committee approved this study.

Information about the training and testing datasets is provided in Table S2. Note that, for the test set of 800 manually labelled images (row 2 in Table S2), initial agreement between the two neuroradiologist labellers was 94.9% (Fleiss' kappa = 0.87), so that a consensus classification decision with a third neuroradiologist was made in 5.1% of cases.

|  | Hospital | Scans | Patients | Age (years) | Abnormal |
|---|---|---|---|---|---|
| **Train** | KCH | 26879 | 20111 | 50.9 ± 16.8 | 69.3% |
|  | GSTT | 16875 | 14245 | 49.8 ± 17.4 | 57.9% |
|  | Pooled | 43754 | 34356 | 50.5 ± 17.1 | 68.1% |
| **Test** | KCH | 400 | 400 | 49.4 ± 15.9 | 67.2% |
|  | GSTT | 400 | 400 | 52.9 ± 16.8 | 47.9% |
|  | Pooled | 800 | 800 | 50.5 ± 17.1 | 57.6% |
| **Simulation** | KCH (2018) | 2986 | 2538 | 51.6 ± 14.3 | 67.1% |
|  | GSTT (2018) | 1875 | 1556 | 50.2 ± 14.8 | 50.1% |

Table S2: Training, testing, and simulation dataset statistics. The patient age distribution is given in terms of (mean ± standard deviation), and 'Abnormal' refers to the fraction of abnormal examinations in each dataset.

### B.0.1. Test set abnormalities

Our test set of 800 scans (400 from KCH, 400 from GSTT) contains the following distinct abnormalities:

**Mass**

- primary intracranial tumour

- haemorrhagic tumour

- epidermoid tumour

- cystic tumour

- Giant perivascular space

- high grade glioma

- low grade glioma

- astrocytoma

- anaplastic oligodendroglioma

- oligodendroglioma

- dysembryoplastic neuroepithelial tumour (DNET)

- multifocal GBM (glioblastoma multiforme)

- cerebral anaplastic lymphoma

- primary lymphoma

- carcinomatous dural infiltration

- meningioma

- chondrosarcoma

- schwannoma

- neoplastic process of internal auditory meatus

- optic nerve mass

- macroadenoma

- subependymoma

- subependymal heterotopia

- transmantle heterotopia

- gangliocytoma

- neurocytoma

- epidermoid tumour

- cerebellar haemangioma

- cerebellar medulloblastoma

- haemangioblastoma

- meningioangiomatosis

- melanosis

- colloid cyst

- choroid fissure cyst

- pituitary cyst

- arachnoid cyst

- neuroglial cyst

- pineal cyst

- neuroepithelial cyst

- abscess

- extra axial lesion

- posterior fossa extra axial lesion

- pineal lesion

    – fourth ventricular nodule

**Vascular**

    – arteriovenous malformation

    – aneurysm

    – cavernoma

    – developmental venous anomaly

**Acute stroke**

    – anterior circulation infarcts

    – posterior circulation infarcts

    – hypoxic ischaemic injury

**Abnormal ventricular configuration**

    – hydrocephalus

    – Chiari I malformation

    – tonsillar ectopia

    – Chiari II malformation

**Haemorrhage**

    – Any acute / subacute haemorrhage e.g., parenchymal, subarachnoid, subdural, extradura

    – Acute microhaemorrhages / petechial haemorrhages

**Extracranial**

    – excessive accumulation of fluid within the mastoid air cells

    – inflammatory change in the paranasal sinuses

**Infective/Inflammatory**

    – encephalitis

    – ventriculitis

    – vasculitis

    – perforating vasculitic process

    – herpes simplex encephalitis

  – Creutzfeldt-Jakob disease

  – pachymeningitis

  – progressive multifocal leukoencephalopathy

  – granulomatous inflammatory processes

  – aspergillus infection

  – toxoplasmosis

  – demyelinating process

  – systemic lupus erythematosus vasculitis

  – Behcet's vasculitis

  – HIV encephalopathy

  – progressive neurosarcoid

**Damage**

  – encephalomalacia

  – wallerian degeneration in brain stem

**Atrophy**

  – hippocampal/medial temporal volume loss in keeping with Alzheimer's dementia

  – generalised brain volume loss/atrophy/involutional change in excess for the patient's age

**Small vessel disease and other ageing related**

  – small vessel disease

  – large perivascular spaces

  – lacunar infarcts

  – cerebral amyloid angiopathy

**Miscellaneous**

  – posterior reversible encephalopathy syndrome

  – osmotic demyelination syndrome

  – pineal calcification

  – polymicrogyria

  – septation at the aqueduct

  – distorted cerebral peduncle

## Appendix C. Retrospective simulation study

To quantify the impact that our model would have in a real clinical environment, we performed a retrospective simulation study using all out-patient examinations performed between 1/1/2018 - 31/12/2018 to simulate what would have happened if our model had been used to suggest the order in which head MRI examinations were reported. We excluded in-patient examinations because at KCH and GSTT in-patient head MRI examinations - which often contain abnormal images - are mandated to be reported on the same day for every day of the year, so that a triage system for these examinations would have little impact.

Each head MRI examination had an associated 'acquisition timestamp' (i.e., the date and time of acquisition) as well as a 'report timestamp' (i.e., the date and time when the radiology report was published by a neuroradiologist for all in the hospital to see), allowing the historical 'report delay' for each examination to be determined. Using ALARM, we then labelled each examination as 'normal' or 'abnormal' in order to stratify report delays by category. We divided the entire one-year observation period into 365 single-day time intervals, and used the same number of examinations which were historically reported in each day as the estimated number of exams which could be feasibly reported from the front of the re-prioritized queue.

The simulation proceeds by stepping through each day and, using the original acquisition timestamp, showing the scans performed on that day to our abnormality classifier model. The model's output (i.e., the predicted image category) was then used to decide where in the reporting queue to insert each image. Note that we use the class (i.e. 'normal' or 'abnormal') rather than the predicted probability to decide where in the queue to insert each scan to avoid easy-to-identify but less urgent abnormalities from jumping ahead of clinically urgent but difficult-to-classify abnormalities in the queue. In this way, the predicted class and time already spent in the queue are used to determine reporting order. Once the day's scans were added to the existing queue, the first N scans at the front of the queue (where N is fixed by the number of scans historically reported that day) were then removed from the front of the queue, and the modelled 'prioritized report delay' (i.e. the difference between the historical 'acquisition time' and our modelled 'report timestamp') for these scans was recorded. At the end of the one-year period, the modelled 'prioritized report delay' for each examination was compared to the historical reporting times. In order to compute p-values, this experiment was repeated 1,000 times under the null hypothesis - that is, assigning a random priority (class) to each image. In this way, we were able to determine the probability that the observed reduction in 'abnormal' reporting times (and concomitant increase in 'normal' reporting times) due to our prioritization system could have occurred by chance.

This simulation was inspired by the work of (Annarumma et al., 2019) which was performed in the context of triaging chest radiographs, and we made use of code which these authors have made available at https://github.com/WMGDataScience/chest_xrays_triaging/blob/master/reporting_delays_simulation/simulate_reporting.py. Our modified code, which is optimised for use with our head MRI scan classifier, is available at https://github.com/MIDIconsortium/Prioritization_simulation.

## Appendix D. Patient and Public Involvement (PPI) survey

We performed Patient and Public Involvement (PPI) meetings and interviews with those with neurological illness, their families and end-users (neuroradiology department personnel).

One question included that normal results would be delayed. Initial PPI with a UK University technology group 'Next Generation Medical Imaging' articulated the following key themes: (1) that patients with abnormalities should 'queue jump' reporting and receive treatment earlier whilst those 'without abnormalities' would accept waiting longer. Subsequent PPI with UK charity members from Brainstrust, Stroke Association, Meningioma UK, Brain Tumour Charity, Tessa Jowell Foundation and Alzheimer's Society PPI strongly agreed with these conclusions. In total 29/30 strongly agreed and 1/30 agreed.

Another theme was that PPI felt that in general they would be guided by the end users (neuroradiologists) assessment of the technology.

The end users strongly agreed (12/12) that patients with abnormalities should 'queue jump' reporting and receive treatment earlier whilst those 'without abnormalities' should wait longer.

# Appendix E. Occlusion sensitivity and error analysis

Impression: Signal change and swelling involving the right amygdala and hippocampus, consistent with a low grade intrinsic neoplasm

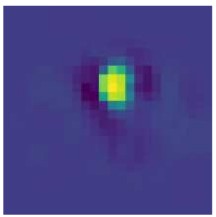 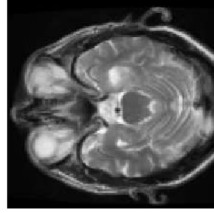

Impression: Right occipital AVM with thrombosed cortical draining vein and associated oedema

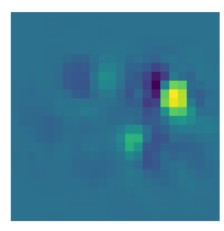 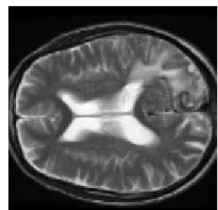

Impression: There is a mature infarct in the right corona radiata.

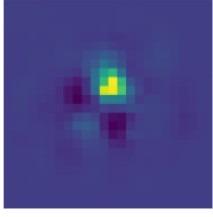 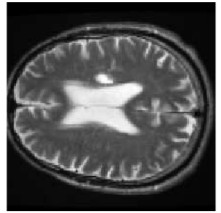

Impression: Evidence of mature left cerebellar damage

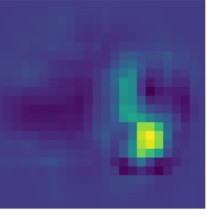 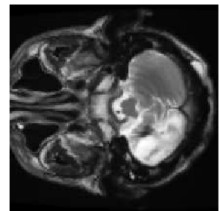

Impression: Cystic intra axial lesion of the left temporal lobe. Differential diagnosis would include gangliocytoma, DNET and also neuroepithelial cyst

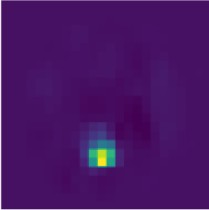 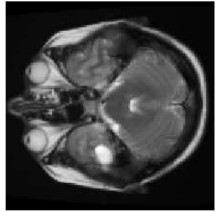

Impression: There is extensive, confluent acute infarction involving the brainstem and both cerebellar hemispheres.

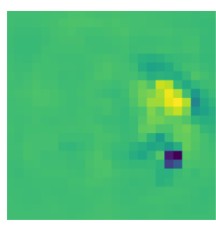 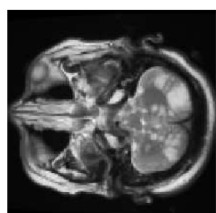

Impression: There is a mass lesion centered on the right middle and inferior cerebellar peduncles with exophytic cystic changes. An intrinsic glial neoplasm appears most likely

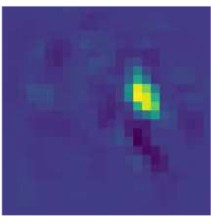 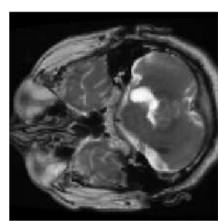

Impression: There is a subdural haematoma overlying the left parietal convexity, mildly indenting the subjacent parietal cortex.

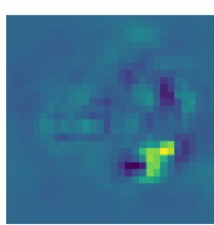 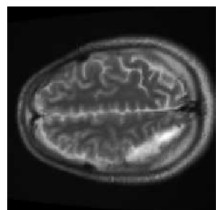

Figure S2: Occlusion sensitivity analysis (kernel size $= 5 \times 5 \times 5$, stride $= 5$) of a representative set of positive (i.e., 'abnormal') images from the combined KCH + GSTT test set. Lighter colors (yellow) correspond to image regions important to model classification, and darker colors (blue) correspond to image regions less important to model classification. For reference, the 'Impression' section of the accompanying radiology reports is shown above each image.

There are a few small areas of high T2 signal in the subcortical frontal white matter bilaterally in keeping with mild small vessel disease, otherwise normal intracranial appearances

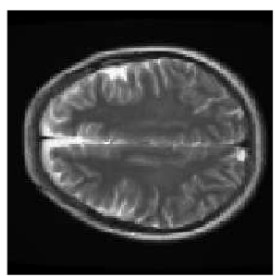

Apart from a few small areas of white matter T2 hyperintensity in keeping with mild small vessel disease there are otherwise normal intracranial appearances.

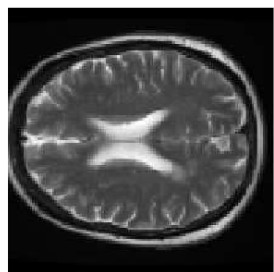

There are areas of patchy subcortical and deep white matter T2 hyperintensity in keeping with mild small vessel disease

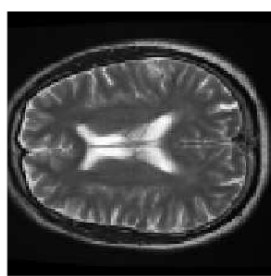

There are several T2 hyperintense foci in the cerebral white matter in keeping with mild to moderate small vessel disease

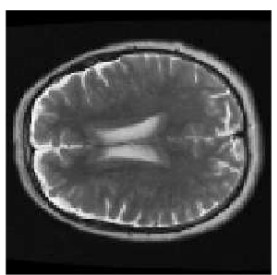

There are numerous focal T2 hyperintensities within the subcortical and deep white matter of the fronto-parietal white matter in keeping with mild to moderate small vessel disease

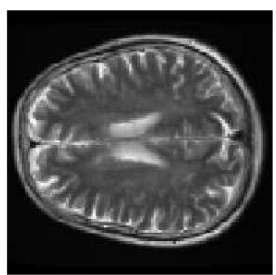

There are multiple T2 hyperintense foci in the periventricular and subcortical white matter of both cerebral hemispheres suggesting mild to moderate small vessel disease

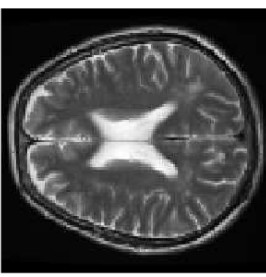

There are extensive abnormal T2 hyperintense foci involving the subcortical, deep and periventricular white matter, in keeping with severe small vessel disease

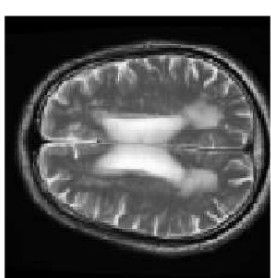

The cerebral white matter and deep grey matter are in keeping with severe small vessel disease

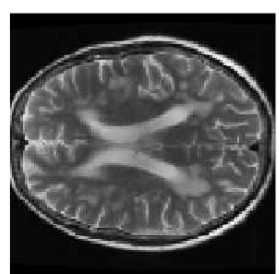

There are foci of T2 hyperintensities involving the periventricular white matter in keeping with severe small vessel disease

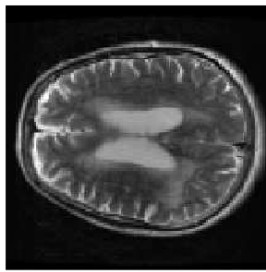

Figure S3: Comparison between 'mild', 'mild to moderate', and 'severe' small vessel disease (SVD). In most cases, our model was able to correctly classify SVD; however, equivocal cases (e.g., 'mild-to-moderate', which had been labelled by our team of neuroradiologists as 'abnormal' to encourage model sensitivity) were sometimes misclassified. Similarly, equivocal cases involving atrophy and enlarged perivascular spaces were sometimes misclassified. Given the degree of subjectivity involved, however, these errors are highly unlikely to have a significant clinical impact.

## Appendix F. Abnormality definitions

Abnormal is defined as one or more abnormality described below. Normal is defined as no abnormality described below.

### F.1. Small vessel disease

(Fazekas et al., 1987) gives a classification system for white matter lesions (WMLs):

1. Mild - punctate WMLS: Fazekas I

2. Moderate - confluent WMLs: Fazekas II

3. Severe - extensive confluent WMLs: Fazekas III

To create a binary categorical variable from this system, if the report was unsure/normal or mild this would be categorized as 'normal' as this never requires treatment for cardiovascular risk factors. However, if there is a description of moderate or severe WMLs, the report would be categorized as 'abnormal' as these cases sometimes require treatment for cardiovascular risk factors.

Included as normal are descriptions of scattered non-specific 'white matter dots' or "foci of signal abnormality" (unless a more defuse or specific pathology is implied) and small vessel disease described as 'minor', 'minimal' or 'modest'.

Conversely, those cases which are described as 'mild to moderate', 'confluent', or 'beginning to confluence' small vessel disease are treated as abnormal.

Genetic small vessel disease, in particular Cerebral Autosomal Dominant Arteriopathy with Subcortical Infarcts and Leukoencephalopathy (CADASIL), is considered abnormal.

### F.2. Mass

– Neoplasms

  – Intra-axial including all primary and secondary neoplasms

  – Extra-axial including all primary and secondary neoplasms (including pituitary adenomas)

– Tumour debulking or partial resection as this implies residual tumour

– Ependymal, subependymal or local meningeal enhancement (non-surgical) in the context of a history of an aggressive infiltrative tumour

– Abscess

– Cysts

  – retrocebellar cyst (mega cisterna magna not included)

  - Arachnoid cysts

  - pineal cysts and choroid fissure cysts

  - Rathke's cleft cysts

- Focal cortical dysplasia, nodular grey matter heterotopia, subependymal nodules and subcortical tubers

- Lipoma

- Chronic subdural haematoma or hygroma (i.e. cerebrospinal fluid (CSF) equivalent)

- Perivascular spaces normal unless giant

- MRI examinations for stereotactic surgical planning alone may have very brief reports. In these scenarios it is typically evident from the clinical information provided that there is a mass e.g., surgical planning for glioblastoma.

Note that findings that typically may have minimal clinical relevance when confirmed by a neuroradiology expert, are included in this category e.g., arachnoid cyst. The rationale is that such a finding might generate a referral to a multidisciplinary team meeting for clarification clinical relevance. We consider that a referral to a multidisciplinary team meeting is a clinical intervention and we aim to ensure that any findings that generate a downstream clinical intervention are included.

### F.3. Vascular

- Aneurysms

    - including coiled aneurysms regardless of whether there is a residual neck or not

- Arteriovenous malformation

- Arteriovenous dural fistula

- Cavernoma

- Capillary telangiectasia

- Old / non-specific microhaemorrhages

- Petechial haemorrhage

- Developmental venous anomaly

- Venous sinus thrombosis

- Vasculitis if associated with vessel changes such as luminal stenosis or vessel wall enhancement

- Arterial occlusion / flow void abnormality or absence

– Venous sinus tumor invasion

– Arterial stenosis. If constitutional / normal variant not included.

Note that findings that typically may have minimal clinical relevance when confirmed by a neuroradiology expert, are included e.g., developmental venous anomaly. The rationale is that such a finding might generate a referral to a multidisciplinary team meeting for clarification of clinical relevance. We consider that a referral to a multidisciplinary team meeting is a clinical intervention and we aim to ensure that any findings that generate a downstream clinical intervention are included.
Note that a finding that might generate a referral to a multidisciplinary meeting for clarification has been within this category e.g. developmental venous anomaly may be ignored in clinical practice, but we included it in the "vascular" granular category.

Examples of vascular-like findings which are considered normal include descriptions of sluggish flow, flow related signal abnormalities (unless they raise the suspicion of thrombus) and vascular fenetrations.

**F.4. Encephalomalacia**

– Gliosis

– Encephalomalacia

– Cavity

– Post-operative tissue changes / appearances are included as encephalomalacia

– Tumour debulking or partial resection as this implies residual tumour

– If the patient has had a craniotomy or biopsy there is likely damage – however, for example in the case of a burr-hole and drain previously inserted into the extra-axial space, this does not automatically constitute damage

– "Post-operative changes / appearances" include as damage

– Tumor debulking or partial resection as this includes cavity plus tumour (labelled as both "damage" and "mass")

– Chronic infarct / sequelae of infarct

– Chronic haemorrhage / sequelae of haemorrhage (with / without haemosiderin staining)

– Cortical laminar necrosis

Encephalomalacia-like findings which are considered normal unless there is a clear description of related parenchymal injury include craniotomy, burr-holes, posterior fossa decompression, and 3rd ventriculostomy.

**F.5. Acute stroke**

– Acute / subacute infarct (if demonstrating restricted diffusion)

  – Include if there are other descriptors indicating a subacute nature such as swelling even though restricted diffusion has normalised

– Parenchymal post-operative restricted diffusion secondary to retraction injury

– Mitochondrial Encephalopathy with Lactic Acidosis and Stroke-like episodes (MELAS) if associated with restricted diffusion

– Hypoxic ischaemic injury if associated with restricted diffusion

– Vasculitis if associated with acute / subacute infarct

**F.6. White matter inflammation**

– Multiple sclerosis (MS) including when some plaques show cavitation

– Other demyelinating lesions including Acute Disseminated Encephalomyelitis (ADEM) and Neuromyelitis Optica spectrum disorder (NMO)

– Inflammatory lesions in Radiologically Isolated Syndrome / Clinically Isolated Syndrome

– Focal cortical thinning i.e., secondary to chronic subcortical / cortical lesions, are labelled as "encephalomalacia" abnormalities

– Progressive Multifocal Leukoencephalopathy (PML)/ Immune Reconstitution Inflammatory Syndrome (IRIS)

– Leukoencephalopathies - congenital or acquired (including toxic)

– Encephalitis / encephalopathy if it involves the white matter, e.g. related to human immunodeficiency virus (HIV) and congenital cytomegalovirus (CMV)

– Posterior Reversible Encephalopathy Syndrome (PRES)

– Osmotic demyelination (central pontine myelinolysis/ extrapontine myelinolysis)

– Susac syndrome

– Radiation if describing white matter abnormality

– White matter changes in the context of vasculitis if clearly attributed to vasculitis.

– Amyloid-related inflammatory change / inflammatory

**F.7. Atrophy**

Volume loss in excess of age is labelled 'abnormal'. Volume loss 'commensurate for age' is labelled 'normal'.

### F.8. Hydrocephalus

– Acute

– Trapped ventricle

– Chronic / stable / improving hydrocephalus (it does not matter whether its compensated or not)

– normal pressure hydrocephalus (NPH)

– Ventricular enlargement

### F.9. Haemorrhage

– Any acute / subacute haemorrhage parenchymal, subarachnoid, subdural, extradural

– Acute microhaemorrhages / petechial haemorrhages

### F.10. Foreign body

– Shunts

– Clips

– Coils

– If significant metalwork is involved in skull repair e.g. in a cranioplasty (or the occasional craniotomy causing extreme intracranial MRI signal distortion)

– If craniotomies are not causing anything other than slight artefact, then these are considered normal

### F.11. Extracranial

– Total mastoid opacification / middle ear effusions

– Complete opacification / obstruction of the paranasal sinuses

– Ignore mucosal thickening

– If there is clearly a well-defined unambiguous polyp then label as abnormal. If it is 'retention cysts' or 'polypoid mucosal thickening' then ignore. If it is something indistinguishable which could be a retention cyst / polyp then ignore. Anything leading to obstruction - always label as abnormal.

– Calvarial / extra-calvarial masses

– Osteo-dural defects

– Encephaloceles

– Pseudomeningoceles

– Extracranial vessel abnormality below the petrous segment e.g. cervical ICA dissection

– Lipoma, Sebaceous cyst or any other mass if extracranial

– Orbital abnormalities (including masses)

  – Including optic nerve pathology affecting the orbital segment of the nerve i.e. meningioma

  – If there is an abnormality of the intracranial segment of the optic nerve /chiasm such as atrophy then label as intracranial misc.

– Cases with tortuous optic nerves with no other features are ignored

– Eye prostheses and proptosis

– Ignore pseudophakia

– Bone abnormality e.g. low bone signal secondary to haemoglobinopathy

– Basilar invagination.

## F.12. Intracranial miscellaneous

All the following are categorized as '1' for Intracranial miscellaneous:

– Cerebellar ectopia

– Brain herniation (for example into a craniectomy defect)

– Clear evidence of idiopathic intracranial hypertension (prominent optic nerve sheaths, intrasellar herniation)

  – Non-specific intrasellar arachnoid herniation / empty sella should be otherwise ignored

  – Non-specific tapering of dural venous sinuses should be ignored

– Spontaneous intracranial hypotension (pituitary enlargement, pachymeningeal thickening, etc)

  – If subdural collections present, these should be also noted separately

– Cerebral oedema or reduced CSF spaces

– Absent or hypoplastic structures such as agenesis of the corpus callosum

– Meningeal thickening or enhancement – for example in the context of neurosarcoid or vasculitis

– Enhancing or thickened cranial nerves

- Infective processes primarily involving the meninges or ependyma (i.e. ventriculitis or meningitis)

- Encephalitis if primarily involves the cortex (HSV/autoimmune encephalitis)

- Excessive or unexpected basal ganglia or parenchymal calcification

- Optic neuritis involving the intracranial segments of the optic nerves or chiasmitis

- Adhesions / webs

- Pneumocephalus

- Colpocephaly

- Superficial siderosis

- Ulegyria

- FASIs / UBOs

- Basal ganglia / thalamic changes in the context of metabolic abnormalities

- Band heterotopia and polymicrogyria

- Hypophysitis

- Seizure related changes

## Appendix G. densenet architecture

Our model uses the DenseNet121 network for visual feature extraction. This network consists of an initial block of 64 convolutional filters (kernel size = [7 x 7 x 7], stride = 2) and a 'max pooling' layer (kernel size = [3 x 3 x 3], stride = 3), followed by four 'densely connected' convolutional blocks. Each dense block consists of alternating point-wise (kernel size = [1 x 1 x 1]) and volumetric (kernel size = [3 x 3 x 3]) convolutions which are repeated 6, 12, 24 and 16 times in the four blocks, respectively. Between each dense block are 'transition layers' which consist of a point convolution (kernel size = [1 x 1 x 1]) and an average pooling layer (kernel size = [2 x 2 x 2], stride = 2). Global average pooling is applied to the output of the $4^{\text{th}}$ dense block, resulting in a 1024-dimension feature vector which, following concatenation with the patient's age, is used for classification.

