# OpenReview forum: "Automated triaging of head MRI examinations using convolutional neural networks"
_MIDL.io/2021/Conference — MIDL 2021_

### Meta-Review · Area_Chair1 · 2021-03-27

**Recommendation:** Accept (Oral & Special Issue Candidate)

**Metareview:**

Four knowledgeable reviewers recommend accept, and keep this recommendation after the rebuttal. All of them agree that the work is impressive regarding the size of the data set and properly evaluated to highlight the soundness of the method. Authors should address the main points in the reviews when preparing a final version.



**Paper Type:**

both

---

### Decision · Program_Chairs · 2021-03-31

**Decision:**

Accept

**Comment:**

Congratulations your paper has been selected as a long oral.